# Kinetics and detectability of the bridgmanite to post-perovskite transformation in the Earth's D″ layer

Christopher Langrand [1], Denis Andrault [2], Stéphanie Durand [3,4], Zuzana Konôpková [5,7], Nadège Hilairet [1], Christine Thomas [3] & Sébastien Merkel [1,6]*

Bridgmanite, the dominant mineral in the Earth's lower mantle, crystallizes in the perovskite structure and transforms into post-perovskite at conditions relevant for the D″ layer. This transformation affects the dynamics of the Earth's lowermost mantle and can explain a range of seismic observations. The thickness over which the two phases coexist, however, can extend over 100 km, casting doubt on the assignment of the observed seismic boundaries. Here, experiments show that the bridgmanite to post-perovskite transition in $(Mg_{0.86},Fe_{0.14})SiO_3$ is fast on geological timescales. The transformation kinetics, however, affects reflection coefficients of $P$ and $S$ waves by more than one order of magnitude. Thick layers of coexisting bridgmanite and post-perovskite can hence be detected using seismic reflections. Morever, the detection and wave period dependence of D″ reflections can be used to constrain significant features of the Earth's lowermost mantle, such as the thickness of the coexistence layer, and obtain information on temperature and grain sizes.

[1] Univ. Lille, CNRS, INRA, ENSCL, UMR 8207 - UMET - Unité Matériaux et Transformations, F-59000 Lille, France. [2] Université Clermont Auvergne, CNRS, IRD, OPGC, Laboratoire Magmas et Volcans, F-63000 Clermont-Ferrand, France. [3] Institute of Geophysics, University of Münster, Corrensstr. 24, 48149 Münster, Germany. [4] Laboratoire de Géologie de Lyon: Terre, Planètes, Environnement, Université de Lyon, École Normale Supérieure de Lyon and CNRS, Villeurbanne, France. [5] DESY Photon Science, Notkestrasse 85, DE-22607 Hamburg, Germany. [6] Institut Universitaire de France, F-75005 Paris, France. [7] Present address: European XFEL GmbH, Holzkoppel 4, 22869 Schenefeld, Germany. *email: sebastien.merkel@univ-lille.fr

The region 200 km above the core-mantle boundary, the D″ layer, is heterogeneous, anisotropic, and acts as a boundary layer for mantle convection[1]. Seismic reflections in D″ have been interpreted as the result of a solid-state phase change[2] and, most frequently, attributed to the transition of the main lower mantle mineral bridgmanite from a perovskite (Pv) to a post-perovskite (pPv) structure[3–6]. The exact nature of the mechanism of the Pv to pPv transition remains largely unknown[7–10] and there is scarce information regarding its kinetics in the literature. Phase transformation kinetics, however, has been shown to strongly affect dynamics in other mantle regions through buoyancy, thermal, and rheological effects[11,12]. The presence and seismic detectability of the Pv-pPv transition in the Earth's mantle also remains a matter of debates as chemical composition strongly affect the thickness and pressure-temperature (P/T) range of the Pv-pPv coexistence layer[13–15]. The interpretation of seismic travel times, however, tends to favor the existence of pPv in D″[16–18]. Some discrepancies still exist; the observed velocity contrast, for instance, is larger than that predicted by a Pv-pPv phase transition[16,19]. Seismic reflection amplitudes and polarities could provide strong additional constrains for deciphering D″ composition and processes[20–22] but their interpretation requires a good understanding of the underlying phase transformations, which is lacking for the case of the Pv to pPv conversion.

Here, we report a high pressure, high temperature (HP/HT) study of Pv to pPv transformations in $(Mg_{0.86},Fe_{0.14})SiO_3$ conducted in the double-sided laser heated diamond anvil cell. The experimental data is used to constrain the kinetics of the Pv to pPv transformation at D″ P/T conditions. We then evaluate the combined effect of both kinetics and thickness of the Pv-pPv coexistence region on the detectability of the Pv-pPv transition using seismic wave reflections in the Earth's mantle. Stresses induced by seismic waves can lead to a re-equilibration process[23,24] and strongly affect the amplitude of the reflected waves. Finally, we suggest new seismic measurements that could help constrain the physical state of the Earth's lowermost mantle.

## Results

**In situ experiments**. A sample of natural enstatite of composition $(Mg_{0.86},Fe_{0.14})SiO_3$ was loaded in a diamond anvil cell inside an argon pressure medium (see Methods). Enstatite was first compressed at ambient temperature to ≈90 GPa and converted to bridgmanite by laser heating to above 2000 K for 30 min while scanning around the sample. Bridgmanite was then compressed to the target pressure (between 116 and 130 GPa, Table 1) at ambient temperature. We then activated the conversion from the Pv structure into pPv by further heating the sample and keeping it at a constant temperature in the 1600–2400 K range. To ensure kinetics data quality, all heating runs with temperature gradients over 100 K between both sample sides or changes in temperature over 100 K over time were discarded. Conversion of Pv to pPv was continuously monitored by collecting X-ray diffraction images every 10 s at the P02.2 beamline of the PETRA III synchrotron (Fig. 1). When no change in X-ray diffraction pattern could be observed, the sample was quenched to room temperature. If possible, the experiment was repeated in another location of the same sample which had not yet been fully converted to pPv. Otherwise, a new sample was used. Out of nine attempts, we successfully studied four different samples, with data at 12 P/T points (Table 1), 2 of which with workable Pv to pPv kinetics data. Measurements at higher temperature were not attempted to allow for reliable time measurement with the beamline setup we used. Sample pressures were determined from the unit cell volumes and the pressure-volume-temperature equations of state (PVT-EOS) of Pv[25] and pPv[26], with a relative uncertainty of ±2 GPa. There are, however, additional uncertainties on absolute pressure due to the inconsistencies in PVT-EOS calibrations[27,28]. This study will hence report two uncertainties for pressure, ±2 GPa on relative pressure (i.e. pressure differences between the reported points) and ±5 GPa on absolute pressures (see Methods). Temperature uncertainty is below 100 K. Resolution on times during data collection is 10 s.

**Kinetics of the Pv to pPv transformation**. A typical sequence of diffraction patterns with the Pv to pPv conversion is shown in Fig. 1. As diffraction peaks of Pv decrease in intensity, the pPv diffraction line intensities increase, indicative of the ongoing transformation (Fig. 2). In some cases, we observe a transformation from Pv to pPv. In others, no transformation to pPv was observed during the time of the experiment (Fig. 3a, Table 1), indicating either that kinetics is slower than the experiment duration or that the P/T conditions are not in the pPv stability domain. For P/T points with partial or full conversion to pPv, the

**Table 1 Conditions for the different experiments.**

| | Sample | P (GPa) | T (K) | n | τ(s) | Starting pPv ratio (%) | Final pPv ratio (%) | V(Å³) Pv | V(Å³) pPv |
|---|---|---|---|---|---|---|---|---|---|
| Pv to pPv | #1 | 129.5 | 1600 | 1.74 | 502 | 7 | 60 | 122.2 | 120.7 |
| | #1 | 129.5 | 1700 | 1.41 | 188 | 12 | 61 | 122.6 | 121.3 |
| | #1 | 129.5 | 1850 | 1.32 | 44 | 48 | 87 | 122.6 | 121.3 |
| | #2 | 126.0 | 1950 | 0.87 | 166 | 21 | 29 | 122.7 | 121.7 |
| | #2 | 126.0 | 2100 | 1.51 | 149 | 0 | 50 | 124.0 | 121.9 |
| | #2 | 126.0 | 2400 | (1.5) | 12 | 33 | 42 | 123.3 | 121.8 |
| Pv | #3 | 116.8 | 1700 | – | 2700 | – | – | 124.2 | – |
| | #4 | 121.0 | 1780 | – | 3600 | – | – | 123.3 | – |
| | #4 | 121.0 | 1620 | – | 1800 | – | – | 123.2 | – |
| | #4 | 121.0 | 1700 | – | 1800 | – | – | 123.3 | – |
| | #4 | 121.0 | 2050 | – | 1800 | – | – | 123.6 | – |
| | #4 | 121.0 | 2250 | – | 1800 | – | – | 123.7 | – |

For runs with successful conversion from Pv to pPv: Avrami exponent (n), characteristic times of the transformation (τ), final weight proportions of pPv, and average unit cell volumes of Pv and pPv. For runs with no conversion to pPv: experiment duration and Pv unit cell volumes. n = 1.5 was imposed in the last fit of the Avrami equation for sample #2. The uncertainties on relative and absolute pressures are ±2 GPa and ±5 GPa, respectively. Temperature uncertainty is below 100 K. Weight proportions from Rietveld analysis are within ±3%. The extracted characteristic transformation times and Avrami exponents have an uncertainty of ±5 s and 0.1, respectively

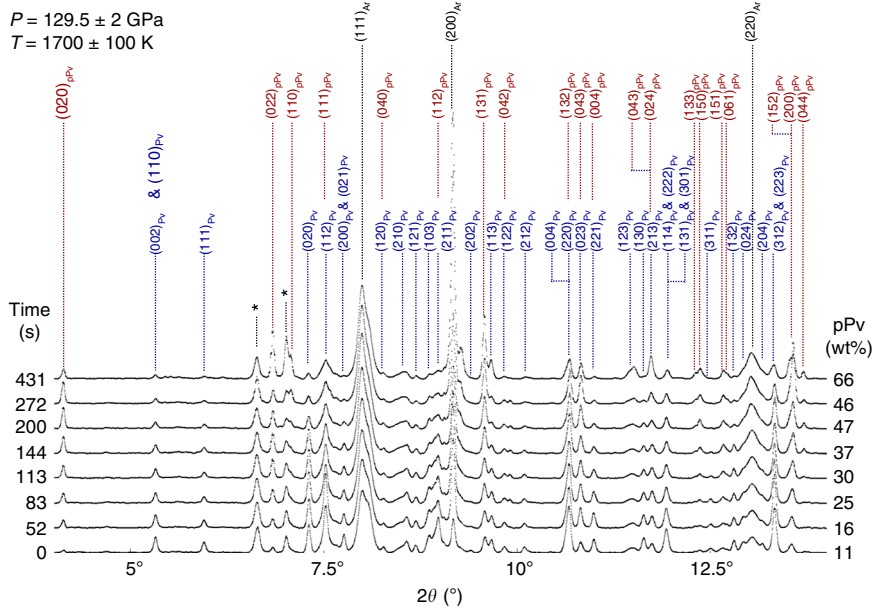

**Fig. 1 Time series of X-ray diffraction data collected in (Mg$_{0.86}$,Fe$_{0.14}$)SiO$_3$ during phase transformation.** Data acquired during the Pv to pPv transformation at 129.5 GPa and 1700 K (Sample #1 in Table 1). Miller indices of most diffraction lines of Pv (blue), pPv (red) and the Argon pressure medium (Ar, black) are indicated on the figure. Stars indicate diffraction lines that are not assigned to either phase and do not change during the transformation. Weight proportions of pPv relative to the (Pv + pPv) assemblage are obtained by a Rietveld refinement with the Maud software[60]. 2$\theta$ angle ranges with Ar peaks were ignored in the refinements.

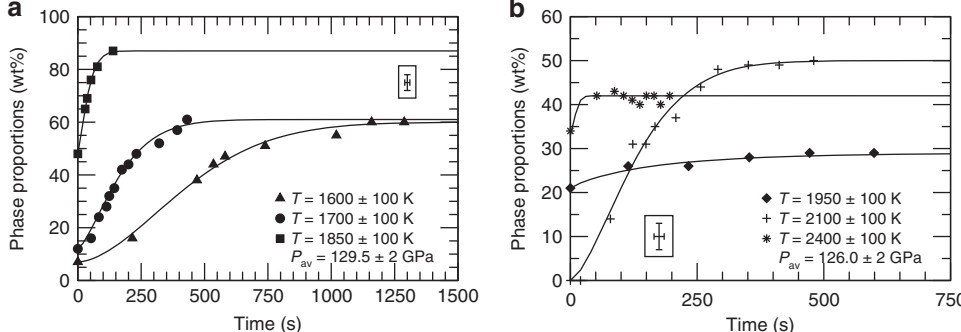

**Fig. 2 Phase transformation vs. time.** pPv weight proportions measured vs. time at **a** 1600 K, 1700 K, 1850 K, and 129.5 GPa and **b** 1950 K, 2100 K, 2400 K, and 126.0 GPa. Symbols are data extracted from the Rietveld refinements. Solid lines are fits of the Avrami model (Table 1). Insets: representative ±1$\sigma$ uncertainties on the experimental data.

resulting proportions vs. time plots (Fig. 2) were analyzed in the framework of the Avrami model[29] with

$$\xi(t) = 1 - e^{-\left(\frac{t}{\tau}\right)^n}, \qquad (1)$$

where $\xi(t)$ is the extend of the transformation, itself deduced from the weight proportion of pPv deduced from the x-ray diffraction data with $\xi = 0$ and $\xi = 1$ for the starting and final weight proportion of pPv, respectively, $n$ the Avrami exponent, and $\tau$ the characteristic transformation time (Table 1, Methods). The Avrami exponent $n$ ranges between 0.87 and 1.74 and, in most cases, is close to 1.5. Such a value for $n$ suggests a mechanism of instantaneous homogeneous nucleation with a diffusion-controlled growth[30]. Previous publications have suggested mechanisms involving homogeneous shear for the Pv-pPv transformation[7,9,10], with contributions of nucleation and growth[8], and this can seem contradictory. It has been shown, however, that transformation mechanisms in silicates can involve both shear and diffusion stages[31,32]. The present result of $n \approx 1.5$

hence indicates that the controlling factor for kinetics is a diffusion-controlled stage in the transformation. This suggested mechanism will have to be reinforced by additional future observations. In the meantime, and in the rest of the analysis, we will consider both mechanisms involving shear and nucleation and growth.

The kinetics data of Table 1 are extrapolated to all D″ P/T conditions using either a nucleation and growth model

$$\tau = \frac{k_2}{T} * \exp\left(\frac{Q_0 - V^*\Delta P}{RT}\right) \left[1 - \exp\left(-\frac{\Delta G}{RT}\right)\right]^{-1}, \qquad (2)$$

or a shear transformation model

$$\tau = k_2' \exp\left(\frac{Q_0}{RT}\right) \left[\sinh\left(\frac{V^*\Delta P}{RT}\right)\right]^{-1}. \qquad (3)$$

$k_2$ and $k_2'$ are constants, $T$ the absolute temperature, $Q_0$ and $V^*$ the absolute activation energy and activation volume of the transformation, $\Delta P$ the overpressure relative to the Pv/pPv phase

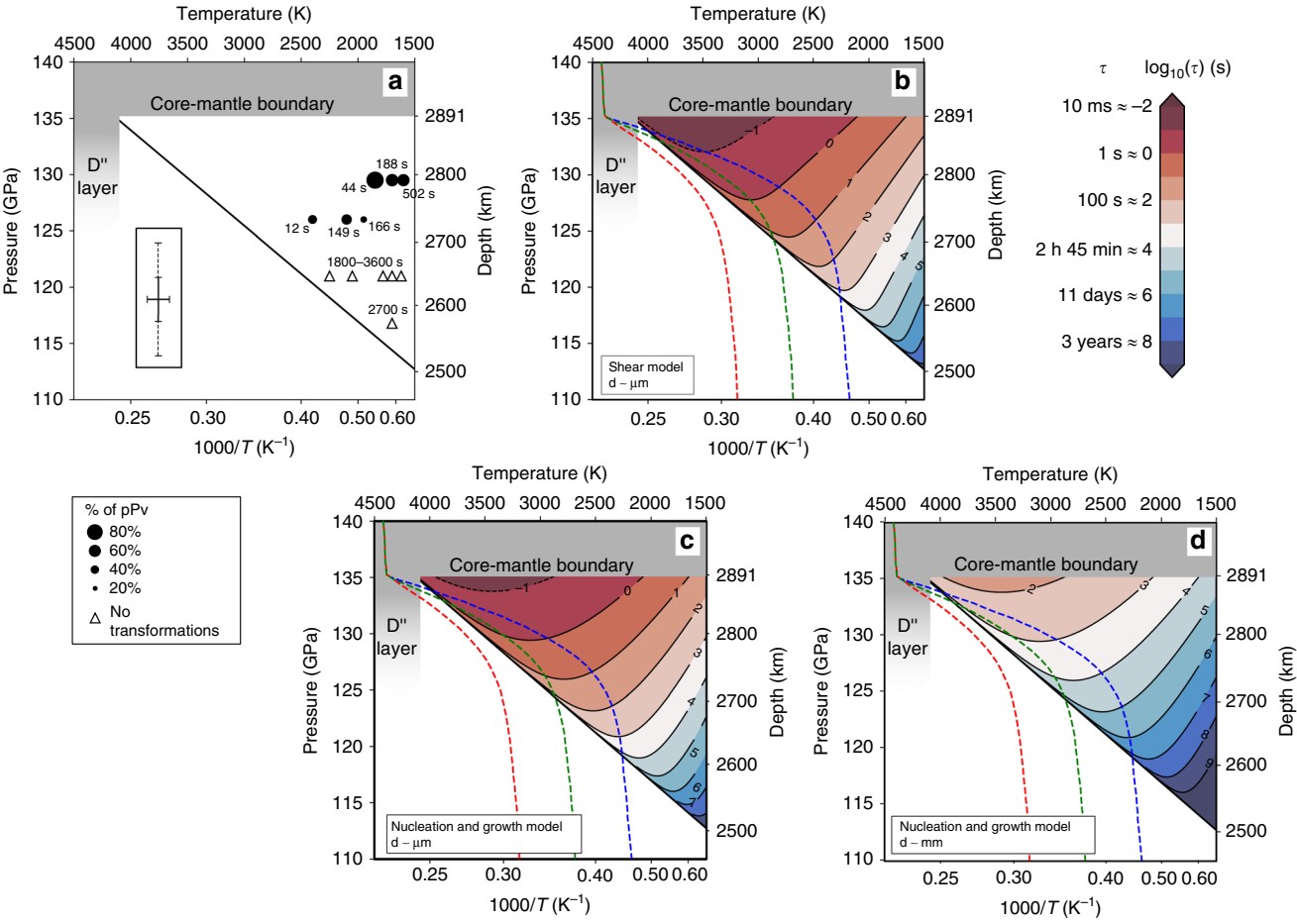

**Fig. 3 Models of bridgmanite transformation kinetics in D″. a** Transformation characteristic times $\tau$ measured in the experiments vs. P and T. The figure also indicates the final weight proportions of pPv by the size of the symbol. Solid line: reference for the Pv/pPv phase boundary (defined by a reference point at $P = 128$ GPa, $T = 3300$ K, and a Clapeyron slope of 8.5 MPa K$^{-1}$). Inset: $\pm 1\sigma$ uncertainties on P and T. Solid and dashed lines represent relative and absolute uncertainties on P, respectively. **b–d** Extrapolated transformation characteristic times vs. P and T for the shear (**b**) or nucleation and growth (**c**, **d**) phase transformations models. Red, green, and blue dashed lines are the hot, warm and cold geotherms of ref. [5]. The shear transformation model is independent of grain size. **d** Extrapolation of the nucleation and growth model to mm grain sizes.

boundary, $R$ the gas constant, and $\Delta G$ the free energy change of the transformation (Fig. 3b, c). Both were adapted from models in the literature[33–35] (see Methods).

The parameters of Eqs. (2) and (3) can be obtained using unconstrained non-linear least squares curve fitting (Methods). The nucleation and growth model leads to $Q_0 = 426 \pm 180$ kJ mol$^{-1}$ and $V^* = 14.4 \pm 6.4$ cm$^3$ mol$^{-1}$ while the shear transformation model leads to $Q_0 = 437 \pm 186$ kJ mol$^{-1}$ and $V^* = 16.1 \pm 6.3$ cm$^3$ mol$^{-1}$, where the expressed uncertainties account for a $\pm 1\sigma$ standard deviation from the non-linear least squares as well as an effect of $\pm 5$ GPa in the location of the Pv/Pv $+$ pPv phase boundary and $\pm 2$ MPa K$^{-1}$ for the corresponding Clapeyron slope. Both models yield very similar results regarding characteristic transformation times at D″ conditions (Fig. 3b, c). The goodness of the fit of kinetics parameters to Eqs. (2) and (3) does not allow to favor one model over another.

Interestingly, for P/T conditions at which we did not observe the Pv/pPv phase transition, both models predict characteristic transformation times above $10^3$ s, which is longer than the duration of our experiments and is, hence, in agreement with our reported non-observations (Table 1). The slow kinetics of the Pv to pPv transformation for temperatures below 2000 K and pressures below 120 GPa is also consistent with reports of

"sluggish" Pv-pPv transformation in the literature and the difficulty of establishing consistent phase diagrams in this P/T region[13–15]. Above 3000 K and 125 GPa, both kinetics models suggest transformation characteristic times below 1 s. With most current HP-HT experimental facilities, such fast kinetics can not be measured and appears instantaneous.

## Discussion

**Upscaling to the D″ layer.** Grain size becomes a critical parameter for scaling experimental kinetics results to the Earth's D″ layer. Diamond anvil cell experiments are performed with grain sizes in the μm range. Grain sizes in D″ are unknown but samples of the upper mantle typically show grain sizes in the mm range.

Shear transformation are driven by the cooperative and homogeneous movement of many atoms. Such transformation (i.e. Eq. (3)) are somewhat independent of grain size. Transformations controlled by diffusion (i.e. Eq. (2)) typically assume a constant product phase growth rate G. A dimensional analysis indicates that growth rate G, is proportional to the ratio of a characteristic length $l$ over a characteristic time $t$. In our problem, the relevant length and time units are the average sample grain size $d$ and transformation characteristic time $\tau$. In a first

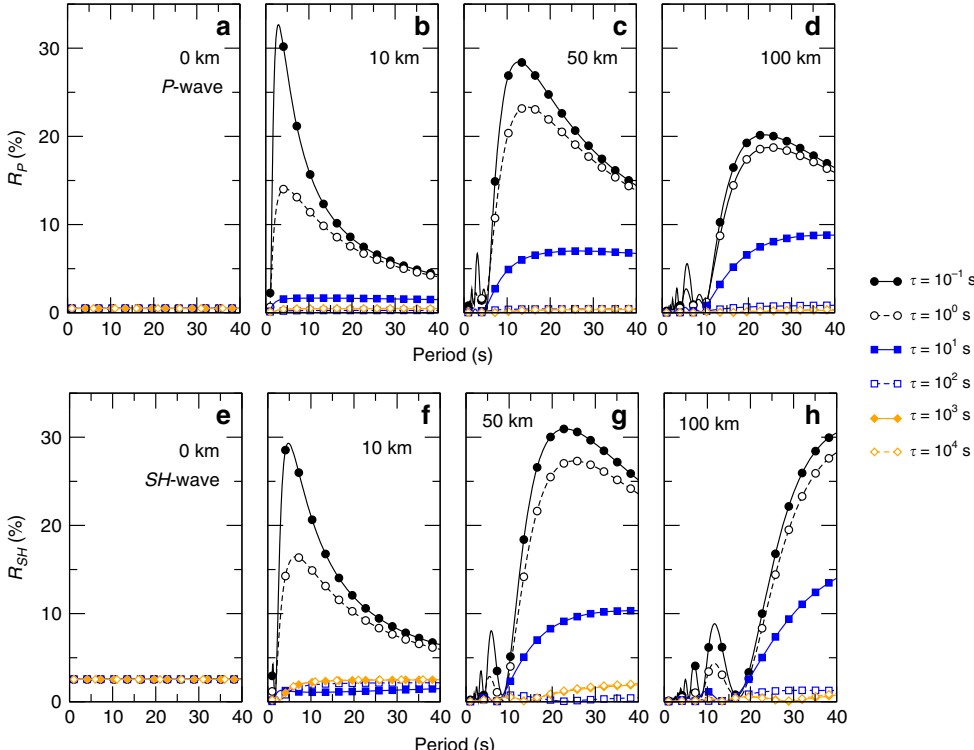

**Fig. 4 Seismic reflections models.** Reflection coefficients for vertically incident P (**a–d**) and SH (**e–h**) waves on a bridgmanite to post-perovskite interface. The figure shows the reflection coefficients as a function of the wavelength of the incident wave for six Pv-pPv transformation characteristic times $\tau$ ranging between $10^{-1}$ and $10^4$ s for an infinitely sharp boundary (**a**, **e**), or a Pv-pPv coexistence layer of 10 km (**b**, **f**), 50 km (**c**, **g**), or 100 km (**d**, **h**). For a sharp boundary, reflection coefficients are 0.5 and 3.5% for P and SH waves, respectively. For thicker Pv-pPv coexistence layers, they can be enhanced or reduced by up to two orders of magnitude depending on transformation kinetics. The wavelength dependence of observable reflections (e.g. >3%) could be used to constrain the thickness of the Pv-pPv coexistence layer.

approximation, we can hence define

$$G \approx k_0 \frac{d}{\tau}, \qquad (4)$$

where $k_0$ is a constant. We can hence expect that, for transformations controlled by diffusion, transformation times scale with grain sizes (Eq. (4)). For experimental grain sizes and D″ P/T conditions, $\tau$ ranges from $10^{-1}$ to $10^4$ s (Fig. 3b, c). For mm grains and diffusion-controlled transformation, we then expect $\tau$ ranging from $10^2$ to $10^7$ s (Fig. 3d), or $10^5$ to $10^{10}$ s (i.e. 300 years) with meter-size grains.

Whatever the pressure, temperature, or grain size, the transformation from Pv to pPv is fast on geological timescales and will not affect long term mantle dynamics. For smaller grain sizes or shear-driven transformations, however, the characteristic times fall in the range of possible interaction with seismic waves. We hence speculate that the Pv to pPv transformation kinetics could be studied with seismic data.

**Detectability of the phase transition.** In regions where two phases coexist, stresses induced by seismic waves propagating through the transition loop can locally disrupt the thermo-dynamically stable system and lead to a re-equilibration process that significantly softens the elastic moduli of the aggregate[23,24]. If the kinetics of the transformation is not instantaneous, it will affect attenuation and reflection coefficients of the seismic waves. We thus propose to use the kinetics timescales determined experimentally to investigate the effect of the Pv-pPv transition on seismic wave reflection amplitudes.

Following the approach used in previous work[22,36], we explore the scenario of vertically incident SH and P waves, with a 1–40 s period, and a D″ interface caused by the Pv to pPv transition using a micro-mechanical model of a phase coexistence loop[24] and the elastic parameters and densities previously computed at 128 GPa and 2800 K[37] (Methods). Following experimental results[15], the Pv/pPv equilibrium coexistence region is assumed to vary over 0–100 km (0–8 GPa). The characteristic transformation times are set to $\tau$ ranging between $10^{-1}$ s and $10^4$ s. In such scenarios, we observe significant variations of the reflection coefficients, ranging from a minimum value below 1% up to 33% (Fig. 4).

The detectability of the Pv-pPv transition using seismic waves can be strongly enhanced, particularly if the Pv-pPv coexistence layer extends over several kilometers and transition times are short, which is plausible for small grain sizes or grain size independent kinetics (Fig. 3). Without the effect of kinetics, a thick Pv-pPv coexistence region would not be detected by seismic waves because of the small amplitudes of reflection, and especially for short period waves, but, with the inclusion of fast kinetics, a gradual transition from Pv to pPv can produce large P and SH reflections. For a 50 km thick interface, for instance, reflection coefficients increase strongly for periods above 5 and 10 s for P and SH waves, respectively, potentially leading to clearly visible seismic reflections.

The experimental results for kinetics were measured in $(Mg_{0.86}, Fe_{0.14})SiO_3$. The addition of other elements such as Ca or Al can affect the timescales of the transformation. The reflection coefficients are weakly affected by a change of density and elastic properties. The results of Fig. 4, however, strongly depend on

kinetics and the thickness of the Pv-pPv coexistence region and both are thoroughly tested in the figures. Figure 4 will hence remain applicable when further experiments quantifying the influence of incorporating other elements on kinetics will become available.

Our calculations only consider vertically incident waves. A variation of the incidence angle would change the reflection coefficients[16]. Other parameters also affect the amplitude of reflected seismic waves such as anisotropy, that induces azimuthal-dependent reflection coefficients[38], or the topography of the reflector, through focusing and defocusing effects[39]. We suggest that the respective effects of kinetics, anisotropy, and topography on D″ reflections could be resolved in a region with a dense distribution of $P$ and $S$-waves reflections, using a range of periods and azimuths. The effect of kinetics is independent of azimuth, unlike that of anisotropy. Topography would generate strongly varying amplitudes of $P$ and $S$-waves within one region, which is not expected for the effect of kinetics.

Our study shows that reflection coefficients can vary over orders of magnitude due to the effect of phase transformation kinetics which could explain previously observed strong variations of D″ reflected waves[16,19]. Our results also show a dependence of the reflection coefficients with the period of the seismic wave. Reflection coefficients for $SH$ waves, for instance, increase strongly above 2, 10, and 20 s for Pv/pPv coexistence region of 10, 50, and 100 km, respectively. As such, the frequency dependence of the detection and amplitude of D″ reflections could be used as a proxy for the existence of a Pv/pPv interface in the deep mantle and constraining the thickness of a potential Pv + pPv coexistence region. Moreover, a measure of the absolute value of the reflection coefficient would constrain the transformation kinetics between both phases. Combined with the results of Fig. 3, such measurement will allow constraining the temperature and grain size in the Earth's D″ layer.

## Methods

**Experimental details.** The starting material was a pure natural enstatite of composition $(Mg_{0.86}, Fe_{0.14})SiO_3$, as determined from electron microprobe analysis. It was loaded with argon pressure medium in diamond anvil cells (DAC) equipped with two beveled diamond anvils with conical support (100–150 μm culet diameter with bevels at 7.5° up to 300 μm diameter). In all cases, sample and pressure medium were contained in a 40–60 μm diameter hole drilled in a rhenium gasket indented to ≈25 μm thickness.

The experiments were performed using monochromatic synchrotron X-ray radiation on beamline P02.2 of the PETRA III synchrotron in Hamburg, Germany[40]. The X-ray beam was focused to ≈2.5 μm both vertically and horizontally, with a wavelength fixed at 0.2918 Å (42.49 keV). Diffraction images were acquired on a Perkin Elmer detector with 2048 × 2048 pixels of 200 × 200 μm² size. The detector to sample distance (551.61 mm), beam center position and detector tilt were determined using a $CeO_2$ standard.

Sample heating was performed with off-axis Yb-Fiber laser heating system, as provided by the beamline (10–30 μm heating spot on sample).

**Data processing.** The collected X-ray diffraction data were integrated and processed using the Rietveld software MAUD[41] in order to extract the cell parameters of Pv, pPv, and the Ar pressure medium along with the weight proportions of all phases (Supplementary Fig. 1). The background was corrected for by using polynomial functions. The fitting did not include the effect of stress nor texture, which is not directly observable in such experiment.

**Pressure and temperature.** Sample temperatures were estimated on both sample sides by pyrometry of the emitted light with a Czerny–Turner spectrograph equipped with a CCD. Temperature uncertainty on each individual measurement is below 30 K but the sample temperature can evolve during the time of a kinetic experiment. As all data points with more than 100 K dispersion in temperature were discarded, the uncertainty on temperature can hence be safely set to ±100 K.

Sample pressure at high temperature was determined from the unit cell volumes and the equations of state of Pv[25] and pPv[26]. We did notice some fluctuations in the exact values of pressure deduced over time during transformation (varying over ±2 GPa at maximum). The fluctuations in cell parameters (leading to an apparent change of pressure) could be due to Fe-Mg chemical exchange between Pv and pPv,

for instance, but could also be an artifact of the Rietveld refinement on a multi-phase recrystallizing sample.

Accounting for those pressure changes in our kinetic model is a priori feasible but introduces numerical instabilities in the rest of the analysis. For each sample, we hence decided to use an average value of pressure determined from either Pv or pPv over the whole duration of the heating, assigning an uncertainty of ±2 GPa for each relative pressure value. This value of ±2 GPa includes the elusive effect of potential Fe-content exchanges between Pv and pPv.

There are, however, additional uncertainties on absolute pressure due to experimental uncertainties and equation of state (EOS) calibrations. Recent publications and reviews[27,28] discuss the details of determining the Pv-pPv absolute transition pressure. At this point, there is no consensus and conflicting results cannot be solely addressed using self-consistent pressure scales nor accounting for effect of the exact sample chemical composition, with up to 10–15 GPa disagreement between studies. The EOS data we used for bridgmanite[25] and their reported uncertainties on parameters leads to an uncertainty of ±3 GPa (±1σ) in our pressure-temperature range. Accounting this systematic error and potential disagreements between EOS, we assign a conservative ±5 GPa on our absolute pressures.

**Fitting of the Avrami model.** Data for pressures and temperatures with successful conversion from Pv to pPv are analyzed in the framework of the Avrami model[29,42,43] (Eq. (1)). The starting, current, and final proportions of pPv define the values for $\xi$ using:

$$\xi = \frac{w - s}{f - s}, \tag{5}$$

where $s$, $f$, and $w$ are the starting, final, and current weight proportion of pPv. Experimental results for weight proportions of pPv vs. time and the corresponding $\xi$ values are listed in Supplementary Tables 1 and 2.

Values for $n$ and $\tau$ are obtained using either a non-linear least-squares fit to Eq. (1) or a linear least-square fit of

$$\ln\left\{\ln\left[\frac{1}{(1-\xi)}\right]\right\} = A\ln(t) + B, \tag{6}$$

where $A$ and $B$ are adjustable variables and $A = n$ and $B = -n\ln\tau$.

During the experiment, we sometimes noticed a counter-intuitive variation of the diffraction peaks intensity with time, resulting is an apparent reversion of the transformation (Supplementary Fig. 2). These are due to heating of the mechanical parts holding the diamond cell in place inducing unavoidable movements of the sample relative to the heating and X-ray diffraction spot. In this case, the sample was brought back to the correct position by adjusting motors below the diamond anvil cell. The corresponding $\xi$ vs $t$ curves were corrected to account for a time offset between both sections of the curves. In the example of Supplementary Fig. 2 the first estimate of $\tau$ is on the order of 498 s. The final result is $\tau = 502$ s. The effect of such sample drifting corrections is minor on the final results.

**Kinetics model.** Characteristic transformation times at constant pressure with successful conversions to pPv are consistent when plotted in an Arrhenius plot, leading to an apparent activation energy of 237 kJ mol⁻¹ (Supplementary Fig. 3a). The data, however, does not fall on the same trend for the different hydrostatic pressures. Such effect of pressure is not accounted for when expressing results using a simple Arrhenius law.

In fact, phase transition kinetics increases both with temperature and distance in pressure from the phase boundary. We hence need to introduce a term of overpressure $\Delta P$, i.e. the distance in pressure from the equilibrium boundary, in addition of temperature in our analysis. Do note that confusion can arise as some kinetics study rely on absolute pressures $P$ rather than overpressure $\Delta P$. Such assumption is correct for phase transitions with a low value of Clapeyron slope but not for other cases.

We tested multiple kinetic laws available in the literature for both nucleation and growth and shear transformation models[30,44–48]. Out of these, two were successful at reproducing all experimental data: the interface-controlled model of refs. [33,34] and the shear model of ref. [35] (Supplementary Fig. 3b, c).

Kinetics equations are typically fitted in terms a growth rate of a product phase vs. physical parameters. Extracting growth rates from experimental data requires an estimation of grain-sizes. Extracting precise information on grain size is not straightforward in in situ diamond anvil cell experiments. We hence rely on the dimensional analysis of Eq. (4) and express our results in terms of characteristic transformation times $\tau$ which can be readily extracted from and compared to the true experimental data.

A law for interface-controlled transformation is typically expressed as[33,34,49]

$$G = k_1 T \exp\left(-\frac{Q_0 - \Delta P V^*}{RT}\right)\left[1 - \exp\left(-\frac{\Delta G_r}{RT}\right)\right] \tag{7}$$

where $k_1$ is a constant, $Q_0$ and $V^*$ the activation energy and volume for growth, $\Delta P$ the overpressure, $\Delta G_r$ the free energy change of the reaction and $R$ the universal gas constant. For the Pv to pPv transformation, $G$ and $\Delta P$ increase as pressure increases away from the Pv-pPv coexistence boundary, hence the minus sign for the definition of a positive activation volume. This expression becomes Eq. (2)

when expressed in terms of characteristic transformation times. $k_2$ in Eq. (2) is a constant that depends on grain size with $k_2 = k_0 d/k_1$, where $k_1$ is the constant in Eq. (7), $k_0$ the constant in Eq. (4), and $d$ the grain size.

Multiple expressions have been suggested for shear transformation models[35]. Most depend on parameters which are unknown for the Pv-pPv transformation, i.e. because they depend on the exact transformation mechanism. Models derived for the calcite $\Longleftrightarrow$ aragonite transition, for instance[35], assume a mechanism similar to that proposed for the Pv to pPv transformation[7,8,10], with a transformation controlled by the development of stacking faults of the daughter phase into the parent phase and kinetics controlled by the motion of partial dislocations associated with these fault. The expression in Eq. (3) is simplified from this model and was found to fit the data reasonably well. In the expression of Eq. (3), $\exp[Q_0/(RT)]$ accounts for the thermal activation and $\sinh[V^*\Delta P/(RT)]$ models the effect of distance from the phase boundary.

**Fitting of kinetics models**. For both the nucleation and growth and shear transformation mechanisms, $k_2$, $Q_0$, and $V^*$ were adjusted to the data of Table 1.

Due to the lack of a proper thermodynamic database for Pv and pPv, and as in previous studies[49], $\Delta G_r$ is estimated from $\Delta P \Delta V$, where $\Delta P$ is the overpressure and $\Delta V$ is the molar volume change of the transformation. $\Delta V$ was determined directly from the X-ray diffraction data. $\Delta P$ was calculated by comparing the pressure difference between the measured P/T point and the Pv/Pv+pPv equilibrium boundary.

We obtain the fitting parameters of Eqs. (2) and (3) using unconstrained non-linear least squares curve fitting, relying on python script and a Levenberg-Marquardt optimization as implemented in the scipy.optimize.curve_fit routine in scipy[50]. The script has been made open-source and can be found online at [https://github.com/smerkel/kinetics-py]. Standard deviation errors on the fit parameters are calculated from the diagonal coefficients of the covariance matrix. Experimental uncertainties in pressure and temperature are not directly accounted for in the Levenberg-Marquardt optimization. They are, however, accounted for indirectly by changing the location of the Pv/Pv + pPv equilibrium boundary (see below).

Calculations of overpressures are sensitive to the location of the Pv/Pv+pPv equilibrium boundary, which is ill-defined for $(Mg,Fe)SiO_3$. Here, we define the Pv/Pv+pPv equilibrium boundary using a reference point and a Clapeyron slope. The reference point, belonging to the Pv/Pv+pPv equilibrium boundary, was extracted from the extrapolation of ref. [14] and is equal to $[P_e; T_e] = [128\text{ GPa}; 3300\text{ K}]$. Estimation of the Clapeyron slope of the Pv/pPv equilibrium boundary range between 5.0 and 11.5 MPa K$^{-1}$[17,13,27,28], somewhat consistent with a seismological estimate of 6 MPa K$^{-1}$[12], with recent measurements leading to 8.5(4) MPa K$^{-1}$[28].

According to our estimate for absolute pressures, we test the effect of a $\pm 5$ GPa error in $P_e$. We also explore Clapeyron slopes of $8.5 \pm 2.0$ MPa K$^{-1}$, which encompasses most publish values in the litterature. Extrapolated transformation characteristic times away from the boundary do not change significantly (Supplementary Figs. 4 and 5), as for the numerical values of $k_2$, $Q_0$, and $V^*$ (Supplementary Table 3). Errors reported for $k_2$, $Q_0$, and $V^*$ in the Results section are the sum of the statistical errors of Supplementary Table 3 and that induced by an error of $\pm 5$ GPa in absolute pressures and $\pm 2$ MPa K$^{-1}$ for the Clapeyron slope.

**Relevance of activation volume and energy values**. The apparent activation energy obtained from the Arrhenius plot for the Pv/(Pv + pPv) transition, 237 kJ mol$^{-1}$, is consistent with other mantle phase transformations[36,49,51–55]. It is, however, significantly lower than that predicted from numerical modeling[9] indicating, probably, differences in the transformation mechanism. The absolute activation energies for the nucleation and growth and shear models refined from Eqs. (5) and (6) are 426 and 437 KJ mol$^{-1}$, respectively, two times higher than the apparent activation energy. The absolute activation energy accounts for the fact experimental measurements were performed away from the equilibrium boundary, which is not taken into account in the apparent activation energies.

The absolute activation energy accounts for the effect of overpressure relative to the phase boundary (e.g. Supplementary Fig. 3b, c) and are hence more appropriate for extrapolations to D″ conditions. The activation volumes for the nucleation and growth and shear models are 14.4 and 16.1 cm$^3$ mol$^{-1}$, respectively. This value is significantly larger than that measured for phase transformations in $(Mg,Fe)_2SiO_4$[36,49]. It is, however, lower than those reported for other transitions such as calcite to aragonite transformation[35].

**Reflection coefficients of vertically incident SH and P waves**. The propagation of the seismic front within a phase loop induces a small perturbation of pressure that can significantly soften the elastic moduli of the aggregate[23]. Hence, body waves crossing a phase loop will experience a strong attenuation that will affect their propagation. This attenuation can be predicted using a mechanical model[24] and depends on the period of the wave relative to the kinetics of the phase transition. The effect is maximum when both are of the same order of magnitude. As shown previously[56], this softening of elastic moduli in the wave front will also affect the reflectivity of the interface.

Reflection coefficients of body waves are computed assuming a phase loop separating two elastic half-spaces divided into multiple sublayers with a linearly increasing proportion of the high pressure phase with depth. Applying Eqs. (13)

and (25) of ref. [24] and using the elastic parameters and densities previously computed at 128 GPa and 2800 K in ref. [37], we obtain the depth-dependence of the elastic moduli through the loop presented in Supplementary Fig. 6. Density is assumed to vary linearly through the loop. Reflection and transmission coefficients are evaluated by propagating elastic waves using the Thomson–Haskell matricial propagator method[57,58] solving for continuity of displacement and stress at each interface. The obtained coefficients depend on the elastic parameters inside the loop as well as the wave frequency and the phase transformation kinetics.

Here, we compute reflection coefficients for vertically incident $SH$ and $P$ waves of period 1 to 40 s. The Pv/pPv coexistence region is set to 0, 10, 50, or 100 km (corresponding to 0–8 GPa) and the characteristic transformation times are set to $\tau$ ranging between $10^{-1}$ s, and $10^4$ s. For a sharp interface (no phase loop between Pv and pPv), reflections coefficients do not depend on kinetics nor on wave period with $R_P = 0.5\%$ and $R_{SH} = 2.5\%$ and can also be compared with results of the Zoeppritz equations[59]. Using a gradient layer and no effect of kinetics, the amplitude of reflected wave should be reduced for thicker Pv-pPv coexistence layers. Due to phase transformation kinetics, however, the reflection coefficients depend on wave period and can be either reduced for slow transition times or enhanced for fast transition times. Results of the calculations are shown in Fig. 4.

Reflection coefficients depend on the angle of incidence. Using the parameters used in this work and the Zoeppritz equations one can show that reflection coefficient will undergo a polarity change, with small reflection coefficients in its vicinity. Modeling the full effects of both kinetics and incidence angle is beyond the scope of this work. As a starting point, we hence assume that the kinetics will modulate the reflection coefficient as shown in Fig. 4. Full calculations of the reflection coefficient, accounting for both kinetics and incidence angle of the $P$ and $SH$ wave will have to be performed in future work to confirm our assumption.

## Data availability
The raw diffraction data used to produce Fig. 1 and the results presented in this paper will be made available by the corresponding author upon reasonable request. The data used to produce Fig. 2 is available in Supplementary Tables 1 and 2. The data used for producing Fig. 3 as well as Supplementary Figs. 3, 4, 5 is in Table 1.

## Code availability
The python code used to fit the kinetics parameters of Eqs. (2) and (3) and generate Fig. 3 and Supplementary Figs. 4 and 5 is available online at [https://github.com/smerkel/kinetics-py]. The program used to generate Fig. 4 will be made available by the corresponding author upon reasonable request.

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

## Acknowledgements

We wish to thank S. Bellayer for assistance with the electron probe analysis and J-P. Perrillat and M. Roskosz for useful discussions. The study was financed from a grant by the Université de Lille and the Région Hauts-de-France for C.L., with support from the Programme National de Planétologie (PNP) of the CNRS, the bilateral ANR-DFG TIMEleSS project (ANR-17-CE31-0025; TH 1530/18-1; SA 2585/3-1), the bilateral PROCOPE Grant n. 40555PC, and the ARCHI-CM project. The project ARCHI-CM of Chevreul Institute (FR 2638) is a grant from the Ministère de l'Enseignement Supérieur et de la Recherche, Région Nord-Pas de Calais and European Regional Development Fund (ERDF). The electron probe microanalysis (EPMA) facility in Lille (France) is supported by the European Regional Development Fund (ERDF). Parts of this research were carried out at the P02.2 beamline of the PETRA III synchrotron at DESY, a member of the Helmholtz Association (HGF). S.D. was supported by DFG grant HAADES DU1634/1-1. We would like to thank H-P. Liermann for assistance during the experiment. The research leading to this result has been supported by the project CALIPSOplus under the Grant Agreement 730872 from the EU Framework Programme for Research and Innovation HORIZON 2020.

## Author contributions

The study was originally designed by S.M. and N.H., with input from D.A. C.L., S.M., D.A., and Z.K. conducted the synchrotron experiment. C.L. processed the experimental data and built the kinetics model with the assistance of S.M., N.H, and D.A. S.D. and C.T. provided the seismic reflection model and input for the relevance of the results for seismic observations. S.M., C.L., and N.H. wrote the paper with input and discussion from all co-authors.

## Competing interests

The authors declare no competing interests
