## [Peer Review File · Nature Communications]

Reviewers' comments:

Reviewer #1 (Remarks to the Author):

I like this paper because of its originality and because it addresses a question long hidden under the carpet: how can D'' layer be seismically detectable if it is believed to be a broad region of coexistence of Pv and pPv? Two comments:

1. Do I understand correctly that the authors ascribe the D'' discontinuity to the "Weidner effect", I.e. fast phase transformation under the action of the seismic wave? Can this really quantitatively explain the observed discontinuity in Vs and nearly zero discontinuity in Vp? These were well explained with properties of Pv and pPv.

2. Small but important correction: the shear mechanism (it is more correct to call it a "plane sliding mechanism") proposed in our work (Oganov et al., 2005) actually implies nucleation-and-growth, but without diffusion. In fact, any equilibrium first-order phase transition must occur via nucleation and growth. Our mechanism can be seen as a physically corrected version of the mechanism of Tsushiya, who proposed pure homogeneous shear.

Reviewer #2 (Remarks to the Author):

The study of Langrand et al. reports on challenging high-pressure experiments on the kinetics of the phase transition from bridgmanite to post-perovskite that is expected to occur in Earth's lowermost mantle. They compare their kinetics data to models of the phase transformation and discuss the implications for the seismic detectability of this transition under lower mantle conditions. Based on this, the authors speculate that frequency-dependent seismic observations of D'' reflections could be used to infer the kinetics of the transition as well as the thickness of the two-phase region.

This work will be important for interpreting seismic observations of the D'' discontinuity in different settings, as the kinetics of the phase transition is generally not considered. It may also help to reconcile observations in previous experiments by providing an explanation for why certain studies do and others do not see a transition at particular pressure and temperature conditions. As such, I think this work is an important contribution to deep Earth research and suitable for publication in Nature Communications. However, I do have some questions regarding the experiments and equilibrium boundary used, as well as some comments on their speculations, that I would like to see addressed before I would recommend that this contribution is accepted for publication.

Below are my comments roughly in order of appearance in the text.

Comments:

1) The authors have tested one composition of $(\text{Mg}_{0.86}, \text{Fe}_{0.14})\text{SiO}_3$ for bridgmanite. How do they expect their results and the kinetics to vary with composition, especially with the presence of Al or Ca?

2) Some questions about the experiments themselves:

i) The authors bring enstatite up to 90 GPa and heat it up to obtain bridgmanite. Why do the authors bring this down to ambient temperature again before heating it up to obtain pPv? Why not keep Br at higher temperature before increasing the pressure?

ii) Line 53 – how did the authors establish that steady state was reached? In some of the experiments, pPv did not form, but the transition is slow as shown by the modeling later on. So how do the authors know there is steady state?

iii) In lines 47 – 48, the authors do not specify the pressures and temperatures. I understand

these are mentioned later on (line 56), but numbers would be useful here.

iv) Line 55, the authors mention 4 samples are studied successfully. Out of how many samples that were attempted? And given that in only 2 of these, the transition is observed, it would be good to mention that here.

3) Line 101-103: I presume these values are obtained as mentioned in lines 123-124 in the Supplementary Material – using a non-linear optimization? It would be good to have this information summarized in the main text as well. Also, how is the optimization done? In the caption of Table S1 it mentions a non-linear least square (method)?

4) Line 143: the authors decide to study transformation times between 10^{-1} s and 10^4 s, while in line 124 they mention that they find transformation times between 1 to 10^4 s. Why do the authors decide to study the smallest transformation times as well, without finding these from their experimental data? Though it is a reasonable choice, it should be justified, given it has implications for the inferred values of the changes in the reflection coefficients and thus the detectability (making the range smaller, particularly at shorter periods).

5) I have a few comments / questions about the uncertainties in pressure.

i) I appreciate the efforts of the authors to quantify the uncertainties in pressure and the effect of these uncertainties on these results (as shown in Fig. S3 as well). However, why do they assign an error of 1GPa in relative pressure, given that they notice that there are fluctuations of 2 GPa during experiments?

ii) How do these uncertainties influence their speculations on the seismic detectability? The total uncertainty in pressure can easily give rise to a 8 GPa difference, which is of the same order as the two-phase region in the mantle they consider.

iii) I also noticed that these tests only address the uncertainty in the equilibrium reference point for the phase boundary. What would be the effect of changing the reference Clapeyron slope that is currently used? Estimates of this slope vary widely and I wonder whether they have any effect on the results? Particularly, what would be the effect of a non-linear Clapeyron slope?

6) When reading lines 138 – 152, I immediately wondered about the effect of topography of the phase boundary, which the authors briefly mention in line 155. Rather than saying that further work is required, I would like to see some discussion on how the authors think that 'in practice' both the kinetics and thickness of the co-existence region could be constrained when there is significant topography as observed by some studies (e.g.). Related to this, how do the authors suggest to separate the effect of a sharpness of the phase transition from the effect of kinetics? I fear that the authors are overselling the potential use of their study here if they cannot indicate ways to resolve all these different aspects (topography, kinetics and two-phase region).

7) Looking at Fig. 3, it is clear that all measurements are at relatively low temperature compared to the model predictions. What is the limitation on higher temperature measurements – are these possible with the current setup? They would be important for verifying the model results and also to constrain the equilibrium boundary more?

I hope the authors can address these comments and will look forward to receiving their revised manuscript.

In addition to the comments above, I have spotted some things that the authors will hopefully not mind correcting, which are detailed below.

1) The authors seem to use a double quote sign for D'' rather than D double-prime: D'' . Related - the phrase in the title in the online system shows up as D' for me, which might be because of this, or because of another reason. I would make sure to have it mention D' in the title.

2) The authors use both bridgmanite (Br) and perovskite (Pv) throughout the text (for example the caption of Fig. 4). I assume that they use bridgmanite when they refer to the mineral and Pv

when they refer to the structure, but for a general audience it would be good to make this distinction clear.

- 3) Line 33 main text: 'somewhat lacking' sounds very vague
- 4) Line 41: 'that could help constraining' should be 'that could help constrain'
- 5) Line 43: mention the composition of enstatite here please
- 6) Line 72: Please add the reference to Avrami here as well besides the Supplementary Information, to clarify this is from other work and there is enough room for more references.
- 7) Line 80: indicates rather than indicate
- 8) Line 85: add 'model' after 'nucleation and growth'
- 9) Line 84 – 87: Please clarify that these models are adapted from the literature, i.e. references [15-18] in the Supplementary Material, given that there is enough room for more references.
- 10) Line 90: 'in terms a' should be 'in terms of a'
- 11) Fig. 1: Could the authors mention the sample number for easy comparison with Table 1? Also, it would be helpful to the reader if the authors could mark the peaks associated with the Br and pPv phase in different colours.
- 12) Fig. 3: Please also mention the Clapeyron slope in the caption, and it would be more helpful to say 6.7 MPa/K rather than 6.7 e-3 GPa/K in the figure itself.
- 13) Line 68 Supplementary Material: I would have thought 'adjusted on the data' should read 'adjusted from the data' instead?
- 14) Line 77 Supplementary Material: 'sections of the curves' instead of 'section of the curves'
- 15) Lines 96 – 105 Supplementary Material: these lines are a repeat of the main text and not required here, given there should not be overlap.
- 16) Line 113 Supplementary Material: would be good to link this to the main text (equation 2).
- 17) Line 117: 'was found to fit the data reasonably well'. Is this a good reason to choose a model? It would be good to have other reasons for choosing it.
- 18) Line 124: what optimization method is used? 'Non-linear least square' as mentioned in the caption of Table S1? Is there a word missing in the caption, e.g. least-squares method?
- 19) Figure S1: the caption mentions 'lost due to a shit of the diamond anvil cell'. I presume this should read 'lost due to a shift of the diamond anvil cell'. Also, the text mentions 'and corrected after 1000 s'. What does this entail?
- 20) Figure S1: Where does the value of 550 s in the panel a) come from?

Reviewer #3 (Remarks to the Author):

This paper presents *in-situ* x-ray diffraction data during bridgmanite to post perovskite transition at the deep lower mantle pressure and temperature conditions. From the data, the authors infer the kinetics of the phase transition, and further discuss the influences of the transition kinetics on seismic wave propagation/reflection at the two-phase coexisting region. The result is original and important to mineral physics. The study has direct impact on broader geophysics, for example, earthquake related studies. It is also of interest to communities beyond geophysics as it shows how Earth's interior works.

The work reported in the paper is technically challenging and data processing is creative, therefore is worthwhile to publish the result in Nature Communications. The following issues, on the other hand, need to be address before publishing.

1. The reported experimental uncertainties on absolute pressure due to the PVT-EOS calibrations might have been underestimated. The uncertainties mainly come from 1) the error in volumetric measurement of the samples (Pv and pPv phase) and 2) uncertainties of the parameters of the PVT-EOS. Since the manuscript does not list the uncertainty of sample volume, readers can only assess the uncertainty from PVT-EOS point of view. In the paper referred in the manuscript, i.e. ref [23] (Wolf et al 2018), $K_0 = 243.8(43)$ GPa, $K'_0 = 4.160(110)$ for

iron-bearing Pv. It means the uncertainty for $K_{>0}$ is larger than 4 GPa, the uncertainty $K_{>0}'$ is larger than 0.1 which may be translate into an uncertainty larger than 10GPa at 100 GPa.

2.Kinetics analysis is the key of the paper. Table 1 shows the fitting result of Avrami exponent (n), characteristic times of the transformation (τ), and final weight proportions of pPv. However, the final wright proportion is not given in the presented Avrami model (Eq. 1). The final weight proportion should be illustrated either at Eq. 1 or Table 1 so that it is easier for readers to reproduce the analyses.

3.Having said above, I assumed Eq. 1 containing the final weight proportion (ξ_f) in the following form $\xi(t) = \xi_f(1 - \exp(-t/\tau)^n)$, and realized that the fit of the Avrami model for $T=1950K$ (solid curve in the bottom of Fig. 2 b) cannot be reproduce. A best fitting for the Avrami model above to the weight proportion data of $T=1950K$ represented in Fig.2b yields $n=0.29$, $\tau=15$ and $\xi_f = 30$. The calculated weight proportions using these parameters and those reported in Table 1 ($n=0.87$, $\tau=166$, $\xi_f = 30$) are shown in the attached figure. The experimental data in the plot is graphically estimated from Fig. 2. There might be large reading errors, but they should not result in such large differences shown in the attached figure, unless the Avrami model containing W_f used here is different from what used in the paper.

4.Although the references are cited, I suggest to include the formula for calculating Reflection Coefficients (shown in Fig. 4) in the supplemental materials to increase the independence of the paper for reading.

5.To demonstrate the robustness of the kinetics data from the in-situ X-ray diffraction, a plot showing Rietveld fitting result is preferred in the section B of supplemental materials.

6.The statement of "transformation mechanisms in silicates can involve both shear and diffusion stages" in the "Kinetics of the Pv to pPv transformation" section can be supported well by the additional reference "Observation of Cation Reordering during the Olivine-Spinel Transition in Fayalite by In Situ Synchrotron X-Ray Diffraction at High Pressure and Temperature" by Chen et al. PRL 2001.

7.Misspelling of bridgmanite ("brigmanite") in the abstract need to be corrected.

Jiuhua Chen

Reviewers' comments:

Reviewer #1 (Remarks to the Author):

I like this paper because of its originality and because it addresses a question long hidden under the carpet: how can D" layer be seismically detectable if it is believed to be a broad region of coexistence of Pv and pPv? Two comments:

1. Do I understand correctly that the authors ascribe the D" discontinuity to the "Weidner effect", I.e. fast phase transformation under the action of the seismic wave?

Yes - The idea is to explore the kinetic time range obtained from the experiment and see how this affects seismic wave amplitudes. The effect of phase transition kinetics on seismic waves has been described by Li & Weidner (2008), but also by Ricard et al. (2009).

The introduction of the paper now includes a citation to both these references and states
Stresses induced by seismic waves can lead to a re-equilibration process^{23,24} and strongly affect the amplitude of the reflected waves.

Can this really quantitatively explain the observed discontinuity in Vs and nearly zero discontinuity in Vp? These were well explained with properties of Pv and pPv.

From mineral physics, the Br-pPv transition in pure MgSiO₃ is sharp and predicts some velocity jump in S waves and a smaller jump (nearly zero) in P-waves. With a more realistic mineralogy, the transition between Br and pPv is wide (e.g. Andrault *et al.*, 2010, Catalli *et al.*, 2009), leading to even smaller reflection amplitudes, especially for short period waves. One way of increasing reflected waves amplitudes is through deformation (Amman *et al.*, 2010 and Thomas *et al.*, 2011) but it will not be sufficient for thick Pv-pPv coexistence layers.

In seismic data, the P-wave and S-wave discontinuities are readily visible (see for example Wysession *et al.*, 1998, AGU Monograph, and Cobden *et al.*, Springer, 2015). The velocity contrast has been estimated to reach up to 3% (and more in few cases) when looking at seismic reflection amplitudes. This would be difficult to be explained with a simple Br-pPv transition.

This manuscript aims at explaining the larger reflection amplitudes observed in seismic studies with the effect on kinetics on phase transitions. The important point of the paper is that the D" discontinuity can be detected even in the presence of a large Pv/pPv coexistence region, thanks to kinetics that enhances the reflection coefficients relative to those of pure Pv and pPv. Reflection coefficients for a pure Pv / pPv interface are significantly lower.

The introduction of the paper now states

The interpretation of seismic travel times, however, tends to favor the existence of pPv in D" ¹⁶⁻¹⁸ but some discrepancies still exist. The observed velocity contrast, for instance, is larger than that predicted by a Pv-pPv phase transition^{16,19}.

With a reference to the papers of Wysession *et al.*, 1998, AGU Monograph, and Cobden *et al.*, Springer, 2015.

As stated above, the introduction now states

Stresses induced by seismic waves can lead to a re-equilibration process^{23,24} and strongly affect the amplitude of the reflected waves.

The abstract already stated

Transformation kinetics is shown to affect reflection coefficients of P and S waves by more than one order of magnitude. Unlike previously thought, thick bridgmanite and post-perovskite coexistence layers could hence be detected using seismic reflections.

2. Small but important correction: the shear mechanism (it is more correct to call it a “plane sliding mechanism”) proposed in our work (Oganov et al., 2005) actually implies nucleation-and-growth, but without diffusion. In fact, any equilibrium first-order phase transition must occur via nucleation and growth. Our mechanism can be seen as a physically corrected version of the mechanism of Tsushiya, who proposed pure homogeneous shear.

This is an important correction. The corresponding section now reads

Previous publications have suggested mechanisms involving homogeneous shear for the Pv-pPv transformation^{7,9,10}, with contributions of nucleation and growth⁸, and this can seem contradictory.

Reviewer #2 (Remarks to the Author):

The study of Langrand et al. reports on challenging high-pressure experiments on the kinetics of the phase transition from bridgmanite to post-perovskite that is expected to occur in Earth's lowermost mantle. They compare their kinetics data to models of the phase transformation and discuss the implications for the seismic detectability of this transition under lower mantle conditions. Based on this, the authors speculate that frequency-dependent seismic observations of D" reflections could be used to infer the kinetics of the transition as well as the thickness of the two-phase region.

This work will be important for interpreting seismic observations of the D" discontinuity in different settings, as the kinetics of the phase transition is generally not considered. It may also help to reconcile observations in previous experiments by providing an explanation for why certain studies do and others do not see a transition at particular pressure and temperature conditions. As such, I think this work is an important contribution to deep Earth research and suitable for publication in Nature Communications. However, I do have some questions regarding the experiments and equilibrium boundary used, as well as some comments on their speculations, that I would like to see addressed before I would recommend that this contribution is accepted for publication.

Below are my comments roughly in order of appearance in the text.

Comments:

1) The authors have tested one composition of (Mg_{0.86},Fe_{0.14})SiO₃ for bridgmanite. How do they expect their results and the kinetics to vary with composition, especially with the presence of Al or Ca?

This is an important question. Al and Ca are known to have an influence on the Pv-pPv transition so one would expect to see an effect for kinetics as well. Answering this question without further experiments would be purely speculative and we would prefer to focus on true results. The effect of kinetics on the amplification of reflection coefficients, however, would be weakly affected by the addition of Al or Ca. The results of Fig. 4 remain mostly valid, whatever the composition. The paragraph at line 178 of the manuscript now starts with

The experimental results for kinetics were measured in $(Mg_{0.86}, Fe_{0.14})SiO_3$. The addition of other elements such as Ca or Al can affect the kinetics timescales of the transformation, although not by orders of magnitude. The enhanced detectability of the Pv-pPv boundary highlighted in Figs. 4 is only weakly dependent on composition, through a change in density and elastic properties, but strongly depends on kinetics and the thickness of the Pv-pPv coexistence region, which are well covered in Figs. 4. These results will hence remain applicable to other compositions that those studied experimentally.

2) Some questions about the experiments themselves:

i) The authors bring enstatite up to 90 GPa and heat it up to obtain bridgmanite. Why do the authors bring this down to ambient temperature again before heating it up to obtain pPv? Why not keep Br at higher temperature before increasing the pressure?

This is hard! Things can go wrong very quickly with laser heating experiments, especially while trying to heat while changing pressure. Moreover, temperature would not be homogeneous over the whole sample (temperature is homogeneous within the size of the x-ray diffraction) and it is more appropriate to perform large pressure steps at ambient temperature.

We did not modify the main text as it may introduce confusion for other readers from the diamond anvil cell community.

ii) Line 53 – how did the authors establish that steady state was reached? In some of the experiments, pPv did not form, but the transition is slow as shown by the modeling later on. So how do the authors know there is steady state?

We stopped the experiment when, within the timeframe of a regular laser heating experiments (i.e. 30 minutes) nothing was happening. Synchrotron beamtime is limited and, without further information, one can not wait for hours with no effect. Laser heating for long durations is also tricky as it can lead to diamond failure. The text was clarified by replacing

After reaching a steady state

with

When no change in X-ray diffraction pattern could be observed

iii) In lines 47 – 48, the authors do not specify the pressures and temperatures. I understand these are mentioned later on (line 56), but numbers would be useful here.

We moved the mention to pressure and temperature at this location. The text now reads

Bridgmanite was then compressed to the target pressure (between 115 and 130 GPa, Table I) at ambient temperature. We then activated the conversion from the Pv structure into pPv by further heating the sample and keeping it at a constant temperature in the 1600-2400 K range.

iv) Line 55, the authors mention 4 samples are studied successfully. Out of how many samples that were attempted? And given that in only 2 of these, the transition is observed, it would be good to mention that here.

The text now reads

Out of 9 attempts, we successfully studied 4 different samples, with data at 12 P/T points (Table I), 2 of which with workable Pv to pPv kinetics data.

3) Line 101-103: I presume these values are obtained as mentioned in lines 123-124 in the Supplementary Material – using a non-linear optimization? It would be good to have this information summarized in the main text as well. Also, how is the optimization done? In the caption of Table S1 it mentions a non-linear least square (method)?

Fits results were obtained using unconstrained non-linear least squares curve fitting, relying on a Levenberg-Marquardt optimisation as implemented in the `scipy.optimize.curve_fit` routine in `scipy`. Standard deviation errors on the fit parameters were calculated from the diagonal coefficients of the covariance matrix.

The main text now includes

Fitting parameters of Eqs. 2 and 3 can be obtained using unconstrained non-linear least squares curve fitting (Supplementary Information).

At lines 113-114.

Section F in the supplements now includes the following paragraph

We obtain the fitting parameters of Eqs. S6 and S7 using unconstrained non-linear least squares curve fitting, relying on python script and a Levenberg-Marquardt optimization as implemented in the `scipy.optimize.curve_fit` routine in `scipy`²⁴. The script has been made open source and can be found online at <https://github.com/smerkel/kinetics-py>. Standard deviation errors on the fit parameters are calculated from the diagonal coefficients of the covariance matrix. Experimental uncertainties in pressure and temperature are not directly accounted for in the Levenberg-Marquardt optimization. They are, however, accounted for indirectly by changing the location of the P_v/P_v+pP_v equilibrium boundary (see below).

4) Line 143: the authors decide to study transformation times between 10^{-1} s and 10^4 s, while in line 124 they mention that they find transformation times between 1 to 10^4 s. Why do the authors decide to study the smallest transformation times as well, without finding these from their experimental data? Though it is a reasonable choice, it should be justified, given it has implications for the inferred values of the changes in the reflection coefficients and thus the detectability (making the range smaller, particularly at shorter periods).

This was an error on our side. 10^{-1} s transformation timescales are possible close to the core-mantle boundary (Fig. 4). The text was modified accordingly in the second paragraph of the “Upscaling to the D” layer” section.

5) I have a few comments / questions about the uncertainties in pressure.

i) I appreciate the efforts of the authors to quantify the uncertainties in pressure and the effect of these uncertainties on these results (as shown in Fig. S3 as well). However, why do they assign an error of 1GPa in relative pressure, given that they notice that there are fluctuations of 2 GPa during experiments?

The standard deviation of the pressures measured during transformations are between 1 and 2 GPa. The text was changed for a safer uncertainty estimate of ± 2 GPa in relative pressures. This uncertainty is not accounted for in the non-linear least square estimate of the kinetics parameters. As explained in the supplements, there could be chemical exchanges between P_v and pP_v with the ongoing transformation that may explain such fluctuations. A detailed evaluation of such process, however, will be an over-interpretation our experimental data.

The paper was adjusted by

- assigning a +/- 2 GPa on relative pressures,
- removing the mention to standard deviation in pressure in the supplements (line 49)

ii) How do these uncertainties influence their speculations on the seismic detectability? The total uncertainty in pressure can easily give rise to a 8 GPa difference, which is of the same order as the two-phase region in the mantle they consider.

The uncertainty in pressure calibration is a long-standing debate in the high pressure research community. The supplementary information now includes a more detailed discussion (see answer to reviewer 3). Solving the issue of pressure calibration in diamond anvil cell experiment is completely out of the scope of this study. In fact, one could argue that the seismic detection of an interface with physical properties in line with those of a Pv/pPv transition should be used to calibrate experimental pressures.

From a mineral physics point of view, one strong conclusion of our study is that the period dependence of the seismic reflection coefficients could be used to as a marker identify the thickness of the two phase regions. If the transition thickness is larger than 100 km (i.e. 8 GPa), the amplification of reflection coefficients due to kinetics will appear at a larger wave period.

We modified the end of the paper to strengthen those points

Our results show a dependence of the reflection coefficients with the period of the seismic wave. Reflection coefficients for SH waves, for instance, increase strongly above 2, 10, and 20 s for Pv/pPv coexistence region of 10, 50, and 100 km, respectively. As such, the frequency dependence of the detection and amplitude of D'' reflections could be used as a proxy for the existence of a Pv/pPv interface in the deep mantle and constraining the thickness of a potential Pv+pPv coexistence region. Moreover, a measure of the absolute value of the reflection coefficient would constrain the transformation kinetics between both phases.

iii) I also noticed that these tests only address the uncertainty in the equilibrium reference point for the phase boundary. What would be the effect of changing the reference Clapeyron slope that is currently used? Estimates of this slope vary widely and I wonder whether they have any effect on the results? Particularly, what would be the effect of a non-linear Clapeyron slope?

This point was fully re-investigated based on recent publications on the topic (see section F in the supplementary materials). We now assign a Clapeyron slope of 8.5 MPa.K⁻¹ rather than 6.7 MPa.K⁻¹ in the previously submitted version and investigate the effect of a +/- 2 MPa.K⁻¹ error in Clapeyron slope. The results are presented in Table S3, Figs S4, and S5.

The main manuscript's corresponding figures (i.e. Fig. 3) has been redrawn for a reference Clapeyron slope of 8.5 MPa.K⁻¹. All numbers for activation energies, activation volume, and transformation constant k_2 in the main manuscript now account for a +/- 5 GPa error in pressure and +/- 2 MPa.K⁻¹ error for the Clapeyron slope.

6) When reading lines 138 – 152, I immediately wondered about the effect of topography of the phase boundary, which the authors briefly mention in line 155. Rather than saying that further work is required, I would like to see some discussion on how the authors think that 'in practice' both the kinetics and thickness of the co-existence region could be constrained when there is significant

topography as observed by some studies (e.g.). Related to this, how do the authors suggest to separate the effect of a sharpness of the phase transition from the effect of kinetics? I fear that the authors are overselling the potential use of their study here if they cannot indicate ways to resolve all these different aspects (topography, kinetics and two-phase region).

Indeed, topography can locally enhance the D'' reflection but one would expect strongly varying amplitudes of P and S-waves within one region (see for example Thomas and Weber, 1997). The manuscript now includes the following suggestions to resolve the effect of kinetics, anisotropy, and topography:

Other parameters also affect the amplitude of reflected seismic waves such as anisotropy, that induces azimuthal-dependent reflection coefficients³⁸, or the topography of the reflector, through focusing and defocusing effects³⁹. We suggest that the respective effects of kinetics, anisotropy, and topography on D'' reflections could be resolved in a region with a dense distribution of P and S-waves reflections, using a range of periods and azimuths. The effect of kinetics is independent of azimuth, unlike that of anisotropy. Topography would generate strongly varying amplitudes of P and S-waves within one region, which is not expected for the effect of kinetics.

7) Looking at Fig. 3, it is clear that all measurements are at relatively low temperature compared to the model predictions. What is the limitation on higher temperature measurements – are these possible with the current setup? They would be important for verifying the model results and also to constrain the equilibrium boundary more?

The measurements of transformation kinetics were not possible with the setup available when we started the project. Kinetics was faster than our measurement techniques. It is now possible with current detector technologies and, even more so, with new X-ray sources such as XFEL in Hamburg. XFEL operations started a year ago and the first diamond anvil cell beamtime are currently underway.

The main text now includes the phrase (in the experiment section)

Measurements at higher temperature were not attempted to allow for reliable time measurement with the beamline setup we used.

I hope the authors can address these comments and will look forward to receiving their revised manuscript.

We hope that we properly answered your comments.

In addition to the comments above, I have spotted some things that the authors will hopefully not mind correcting, which are detailed below.

1) The authors seem to use a double quote sign for D'' rather than DD double-prime: D''. Related - the phrase in the title in the online system shows up as D' for me, which might be because of this, or because of another reason. I would make sure to have it mention D' in the title.

The paper is written in latex, and we wrote D'' as $D^{\prime\prime}$. The online system might have removed one of the prime symbols during submission but we have no control over this. We will have to pay attention to the final version to make sure those bugs are resolved.

2) The authors use both bridgmanite (Br) and perovskite (Pv) throughout the text (for example the caption of Fig. 4). I assume that they use bridgmanite when they refer to the mineral and Pv when they refer to the structure, but for a general audience it would be good to make this distinction clear.

The introduction already states “... transition of the main lower mantle mineral bridgmanite from a perovskite (Pv) to a post-perovskite (pPv) structure ...”

We also modified the text in the “In situ experiments” section to read
... we activated the conversion from the Pv structure into pPv by further heating the sample ...

3) Line 33 main text: ‘somewhat lacking’ sounds very vague

The word *somewhat* was removed.

4) Line 41: ‘that could help constraining’ should be ‘that could help constrain’

Done

5) Line 43: mention the composition of enstatite here please

Done

6) Line 72: Please add the reference to Avrami here as well besides the Supplementary Information, to clarify this is from other work and there is enough room for more references.

Done

7) Line 80: indicates rather than indicate

Done

8) Line 85: add ‘model’ after ‘nucleation and growth’

Done

9) Line 84 – 87: Please clarify that these models are adapted from the literature, i.e. references [15-18] in the Supplementary Material, given that there is enough room for more references.

Done

10) Line 90: ‘in terms a’ should be ‘in terms of a’

Done

11) Fig. 1: Could the authors mention the sample number for easy comparison with Table 1? Also, it would be helpful to the reader if the authors could mark the peaks associated with the Br and pPv phase in different colours.

Done

12) Fig. 3: Please also mention the Clapeyron slope in the caption, and it would be more helpful to say 6.7 MPa/K rather than 6.7 e-3 GPa/K in the figure itself.

The location of the reference line for the $P_v \rightarrow P_v + pP_v$ transformation is now described in the figure caption, both for the reference P/T point and the Clapeyron slope.

13) Line 68 Supplementary Material: I would have thought ‘adjusted on the data’ should read ‘adjusted from the data’ instead?

Done

14) Line 77 Supplementary Material: ‘sections of the curves’ instead of ‘section of the curves’

Done

15) Lines 96 – 105 Supplementary Material: these lines are a repeat of the main text and not required here, given there should not be overlap.

It is true that there is a repetition over a few lines but we would rather keep a self-consistent section. Readers interested in the details of our analysis will not have to go back and fourth between the main paper and the supplement.

16) Line 113 Supplementary Material: would be good to link this to the main text (equation 2).

Done

17) Line 117: ‘was found to fit the data reasonably well’. Is this a good reason to choose a model? It would be good to have other reasons for choosing it.

The text now reads

Models derived for the calcite \rightleftharpoons aragonite transition, for instance¹⁹, assume a mechanism similar to that proposed for the P_v to pP_v transformation²¹⁻²³, with a transformation controlled by the development of stacking faults of the daughter phase into the parent phase and kinetics controlled by the motion of partial dislocations associated with these fault. The following expression, simplified from this model, was found to fit the data reasonably well:

18) Line 124: what optimization method is used? ‘Non-linear least square’ as mentioned in the caption of Table S1? Is there a word missing in the caption, e.g. least-squares method?

Section F in the supplement now reads

We obtain the fitting parameters of Eqs. S6 and S7 using unconstrained non-linear least squares curve fitting, relying on python script and a Levenberg-Marquardt optimization as implemented in the `scipy.optimize.curve fit` routine in `scipy`²⁴. The script has been made open source and can be found online at <https://github.com/smerkel/kinetics-py>. Standard deviation errors on the fit parameters are calculated from the diagonal coefficients of the covariance matrix.

Reference to the non-linear least square method was removed from Table S1 as the fitting procedure is now well described in the text.

19) Figure S1: the caption mentions ‘lost due to a shit of the diamond anvil cell’. I presume this should read ‘lost due to a shift of the diamond anvil cell’. Also, the text mentions ‘and corrected after 1000 s’. What does this entail?

Indeed, there was an f missing, we are talking about a *shift of the diamond anvil cell*. The correction was made by moving the sample (physically). The text now reads

The sample alignment with lasers and x-rays is lost due to a shift of the diamond anvil cell after 600 s and corrected after 1000 s by moving the sample back in front of the x-ray and laser beams.

20) Figure S1: Where does the value of 550 s in the panel a) come from?

550 s is the fitted delay between both portions of the curve. This is now explained in the caption of Fig. S2.

Reviewer #3 (Remarks to the Author):

This paper presents in-situ x-ray diffraction data during bridgmanite to post perovskite transition at the deep lower mantle pressure and temperature conditions. From the data, the authors infer the kinetics of the phase transition, and further discuss the influences of the transition kinetics on seismic wave propagation/reflection at the two-phase coexisting region. The result is original and important to mineral physics. The study has direct impact on broader geophysics, for example, earthquake related studies. It is also of interest to communities beyond geophysics as it shows how Earth's interior works.

The work reported in the paper is technically challenging and data processing is creative, therefore is worthwhile to publish the result in Nature Communications. The following issues, on the other hand, need to be address before publishing.

1. The reported experimental uncertainties on absolute pressure due to the PVT-EOS calibrations might have been underestimated. The uncertainties mainly come from 1) the error in volumetric measurement of the samples (Pv and pPv phase) and 2) uncertainties of the parameters of the PVT-EOS. Since the manuscript does not list the uncertainty of sample volume, readers can only assess the uncertainty from PVT-EOS point of view. In the paper referred in the manuscript, i.e. ref [23] (Wolf et al 2018), $K_0=243.8(43)$ GPa, $K_0'=4.160(110)$ for iron-bearing Pv. It means the uncertainty for K_0 is larger than 4 GPa, the uncertainty K_0' is larger than 0.1 which may be translate into an uncertainty larger than 10GPa at 100 GPa.

As answered to reviewed 2, uncertainties in absolute pressures are difficult to fully estimate in diamond anvil cell experiments.

The reviewer slightly overestimated the error based on the EOS of Wolf *et al*, 2018. A simple Birch-Murnaghan EOS at ambient T with v/v_0 of 0.75 leads to P ranging between 121 and 128 GPa with K_0 ranging between 239.5 and 248.1 and K'_0 ranging between 4.06 and 4.26 and this includes the worst-case scenarios. A full error propagation calculation based on the EOS and 1σ uncertainties of Wolf *et al*, 2018 leads to an uncertainty on the order of +/- 3 GPa at ~130 GPa and 1500 K.

Recent publication (Sun *et al*, 2018) also estimate a 10-15 GPa disagreement between studies on the Pv to pPv transformation, stating that conflicting results cannot be solely addressed using self-consistent pressure scales nor accounting for effect of the exact sample chemical composition.

We hence assigned a conservative +/-5 GPa on our absolute pressures.

The text in the supplement (section C) now reads

Recent publications and reviews^{6,7} discuss the details of determining the Pv-pPv absolute transition pressure. At this point, there is no consensus and conflicting results cannot be solely addressed using self-consistent pressure scales nor accounting for effect of the exact sample chemical composition, with up to 10-15 GPa disagreement between studies. The equation of state (EOS) data we used for bridgmanite³ and their reported uncertainties on parameters leads to an uncertainty of ± 3 GPa ($\pm 1\sigma$) in our pressure-temperature range. Accounting this systematic error and potential disagreements between EOS, we assign a conservative ± 5 GPa on our absolute pressures.

2. Kinetics analysis is the key of the paper. Table 1 shows the fitting result of Avrami exponent (n), characteristic times of the transformation (τ), and final weight proportions of pPv. However, the final weight proportion is not given in the presented Avrami model (Eq. 1). The final weight proportion should be illustrated either at Eq. 1 or Table 1 so that it is easier for readers to reproduce the analyses.

Table 1 now includes the starting and final weight proportion of pPv for each transformation. The text of the main manuscript now says that

$\xi(t)$ is the extend of the transformation, itself deduced from the weight proportion of pPv deduced from the x-ray diffraction data with $\xi = 0$ and $\xi = 1$ for the starting and final weight proportion of pPv, respectively

The supplements now states that

The starting, current, and final proportions of pPv define the values for ξ using:

$$\xi = (w - s) / (f - s),$$

where s , f , and w are the starting, final, and current weight proportion of pPv. Experimental results for weight proportions of pPv vs. time and the corresponding ξ values are listed in Table S1 and S2.

All results regarding phase proportions and ξ vs. time are now listed in Table S1 and S2.

3. Having said above, I assumed Eq. 1 containing the final weight proportion (ξ_f) in the following form $\xi(t) = \xi_f(1 - \exp(-t/\tau))^n$, and realized that the fit of the Avrami model for $T=1950$ K (solid curve in the bottom of Fig. 2 b) cannot be reproduced. A best fitting for the Avrami model above to the weight proportion data of $T=1950$ K represented in Fig. 2b yields $n=0.29$, $\tau=15$ and $\xi_f= 30$. The calculated weight proportions using these parameters and those reported in Table 1 ($n=0.87$, $\tau=166$, $\xi_f= 30$) are shown in the attached figure. The experimental data in the plot is graphically estimated from Fig. 2. There might be large reading errors, but they should not result in such large differences shown in the attached figure, unless the Avrami model containing W_f used here is different from what used in the paper.

The reviewer did not account for a non-zero weight proportion of pPv before the start of the measurements, which was not clear in the submitted version of the paper. The answer to question 3 and the addition of supplementary Tables S1 and S2 should clarify this point.

There are uncertainties and rounding effects in the fit of the Avrami model that can lead to a few seconds shift in the fitting results but this will have no effect on the global kinetics model for which transformation times vary over orders of magnitude.

4. Although the references are cited, I suggest to include the formula for calculating Reflection Coefficients (shown in Fig. 4) in the supplemental materials to increase the independence of the paper for reading.

The main paper now states

Following the approach used in previous work^{22,36}, we explore the scenario of vertically incident SH and P waves, with a 1 to 40 s period, and a D'' interface caused by the Pv to pPv transition using a micro-mechanical model of a phase coexistence loop²⁴ and the elastic parameters and densities previously computed at 128 GPa and 2800 K³⁷.

along with the proper references.

5.To demonstrate the robustness of the kinetics data from the in-situ X-ray diffraction, a plot showing Rietveld fitting result is preferred in the section B of supplemental materials.

We now show a sample Rietveld fit in Fig. S1.

6.The statement of “transformation mechanisms in silicates can involve both shear and diffusion stages” in the "Kinetics of the Pv to pPv transformation" section can be supported well by the additional reference “Observation of Cation Reordering during the Olivine-Spinel Transition in Fayalite by In Situ Synchrotron X-Ray Diffraction at High Pressure and Temperature” by Chen et al. PRL 2001.

Thank you for this very relevant suggestion. The paper is now cited in the manuscript.

7.Misspelling of bridgmanite (“brigmanite”) in the abstract need to be corrected.

Done.

Jiuhua Chen

REVIEWERS' COMMENTS:

Reviewer #1 (Remarks to the Author):

I am satisfied with the revised manuscript and find it ready for publication

Reviewer #3 (Remarks to the Author):

Kinetics works even in the deep mantle where geological scale dominant. This is what the paper shows to the readers. The previous comments are well addressed. Support for publishing the original work, and suggest to Editor's Highlight